# Examining the Role of Source Evaluation in Athlete Advocacy: How Can Advocate Athletes Inspire Public Involvement in Racial Issues?

**Waku Ogiso** [1,]*, **Hiroaki Funahashi** [2] and **Yoshiyuki Mano** [3]

1   Graduate School of Sport Sciences, Waseda University, Nishitokyo 202-0021, Japan
2   School of Health and Sport Sciences, Chukyo University, Toyota 470-0393, Japan
3   Faculty of Sport Sciences, Waseda University, Nishitokyo 202-0021, Japan
*   Correspondence: wakuogiso@akane.waseda.jp; Tel.: +81-42-461-1241

**Abstract:** Athlete advocacy is recognized as an important method of persuading the public on social issues, and it demonstrates the role of athletes in achieving racial justice. However, how athlete advocacy can gain the persuasiveness to encourage public involvement remains unclear. This study investigates how the evaluation of an advocate athlete functions to encourage public issue involvement, focusing on Naomi Osaka's racial advocacy. In particular, driven by balance theory and attribution theory, this study examines the effects of five sociopsychological factors on public involvement in racial issues: perceived credibility, hypocrisy, cause fit, effort expended, and role model status of advocate athletes. Data were collected from a cross-sectional online survey of 855 Japanese adults who were aware of Osaka's advocacy. The findings highlight that public involvement in racial issues is significantly associated with the evaluations of the athlete's credibility and hypocrisy. These evaluations are further influenced by perceptions of the athlete's cause fit and role model status. This study enriches the literature on the management of sports for social change by demonstrating the importance of source evaluation in athlete advocacy in achieving advocacy outcomes. Our evidence implies that athletes looking to promote racial justice issues should effectively be seen as credible, knowledgeable, and non-hypocritical in their issue advocacy.

**Keywords:** athlete advocacy; social influence; public involvement; Black Lives Matter; racial justice; interpersonal persuasion

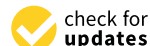



## 1. Introduction

Athlete advocacy represents efforts by elite athletes to advocate for social and political issues; it is currently gaining attention among sport management and sociology researchers (e.g., Agyemang et al. 2020; Cunningham et al. 2021; Schmidt et al. 2019; Yan et al. 2018). Prominent elite athletes, such as Colin Kaepernick, LeBron James, and Megan Rapinoe, to name a few, have advocated for social reforms in matters, such as racism, social inequality, and injustice (Agyemang et al. 2020; Evans et al. 2020). Among others, the Black Lives Matter (BLM) movement, which made significant strides in 2020, has brought public attention to the influence of athletes in achieving racial justice.

Athlete advocacy aims to leverage the fame and persuasiveness of athletes to raise public awareness and engagement on social issues (Agyemang et al. 2020; Kaufman 2008). In other words, a key outcome of athlete advocacy is public involvement. Public involvement is conceptualized as an individual's cognitive, emotional, and behavioral engagement with social issues (Rothschild 1984). Athlete advocacy, according to some academics, may serve as a meaningful vehicle for affecting attitudes and behaviors, such as public awareness, knowledge, and information-seeking habits on the specific issues being advocated (e.g., Babiak et al. 2012; Casey et al. 2003; Frederick et al. 2019; Nownes 2021). Accordingly, it is

necessary to deepen the discussion on how advocate athletes can use their persuasiveness to enhance public involvement.

Although the growing body of literature has shown the effectiveness of athlete advocacy in society, how athlete advocacy gains persuasiveness remains unclear. In particular, how the receivers of athletes' persuasion evaluate and accept their message has not been adequately discussed. Although all athletes should not be expected and forced to engage in social issue advocacy (Coombs and Cassilo 2017), advocate athletes, referring to athletes who choose to advocate social issues, can be agents for disseminating social and political issues (Agyemang et al. 2020; Kaufman 2008; Pelak 2005). Based on this ability, it is necessary to discuss how receivers' evaluations of advocate athletes function, which is an indispensable part of persuasion (Hovland et al. 1953). Therefore, this study examines the role of the receivers' evaluations of advocate athletes in encouraging public issue involvement through the lens of persuasion.

We focus on Naomi Osaka, a notable example of an advocate athlete. Osaka is a multiracial professional tennis player, who supported the BLM movement in 2020. During the 2020 US Open, she wore a different mask before each of her seven games in the tournament; each mask featured the name of a Black victim of racial injustice. Her activity was intended to convey the seriousness of racial issues to a broad audience, including the Japanese public (Deflem 2022a; Ramsay 2020). Osaka's racial advocacy targeting the Japanese can be a suitable case to examine the relationship between source evaluation and public issue involvement from the following three perspectives: (1) the Japanese public is generally regarded as insensitive to and unfamiliar with racial issues owing to their ethnically homogeneous society (Iwabuchi and Takezawa 2015), (2) Osaka's advocacy is an unprecedented athlete's racial advocacy in Japan, and (3) Osaka has received both praise and criticism for her advocacy. Thus, this study centers on Osaka as the research focus in Japan and investigates the function of public evaluation in her advocacy. Note that instead of assessing Osaka's persuasiveness, the present study intends to examine the relationship between the receivers' evaluations of Osaka as an advocate and involvement in public issues.

The present study provides potential theoretical contributions to the sport management and sociology literature by answering the question: "How do the evaluations of advocate athletes function in fostering public issue involvement?" Consistent with the academic trend of managing sport for social change and justice (Cunningham et al. 2021; Love et al. 2019, 2021; Sherry et al. 2015), we demonstrate how the reputations of athletes advocating for social issues work from the perspective of the receivers of advocate athletes' messages. Cunningham et al. (2021, p. 31) expressed the view that "the recognition that sport and athletes can play a role in shaping cultural discourse and promoting social justice" is largely missing from the sport management research. Given the importance of attitudinal and behavioral changes at the individual level in reducing racial discrimination (Pager and Shepherd 2008), exploring the psychological process underlying the acceptance of athletes' messages would help further define the role of advocate athletes as communicators for achieving racial justice.

## 2. Theoretical Framework and Literature Review

### 2.1. Existing Logic behind Potential Effects of Athlete Advocacy on Public Issue Involvement

Celebrity athletes are often compelling advocates of social and political issues because they can attract public attention, allowing others to observe their beliefs and actions (Babiak et al. 2012; Kim and Na 2007). This idea is underpinned by the human capacity for observational learning, which means that individuals expand their beliefs and behaviors by learning through the information they receive (Bandura 2001). This process could be facilitated by the societal power of sport and athletes. To elaborate, Smith and Westerbeek (2007, p. 25) argued that "sport, more than any other potential vehicle, contains qualities that make it a powerful force in effecting positive social contributions". Athletes have unique resources, such as fame and influence, which they can leverage to effectively

advocate for social and political issues (Babiak et al. 2012). For example, Nownes (2021) demonstrated that athletes' messages about health issues impact the public's awareness of these issues. Additionally, athletes are often expected to be role models of behaviors and values (Brown and de Matviuk 2010; Bush et al. 2004); in particular, during the BLM movement, athletes were identified as possessing unique perspectives and exercising significant influence (Nielsen 2020). Additionally, Ogiso et al. (forthcoming) observed that exposure to athletes' racial advocacies is associated with the level of individuals' issue involvement, using the Japanese sample. Based on the discussion above, exposure to the advocate athletes' messages has the potential to enhance the public's psychological and behavioral involvement due to their status and fame.

### 2.2. Balance Theory and Attribution Theory

Athlete advocacy comprises the dissemination of messages by athletes that aim to foster public issue involvement, which has the potential to drive proactive civic participation, such as encouraging donations, volunteering, and social activism (Babiak and Sant 2021). This study adopts two theoretical foundations to understand interpersonal persuasion. First, to understand how public issue involvement (i.e., the outcome of athlete advocacy) is influenced by an individual's evaluation of an advocate athlete, this study adopts the balance theory (Heider 1946). Furthermore, attribution theory (Heider 1958; Kelley 1973) is employed to interpret how individuals formulate source evaluation in persuasive communication.

Balance theory (Heider 1946) helps explain how individuals accept or reject a specific persuasion from an interpersonal relationship perspective. This theory posits that, in triadic relationships, humans prefer a balanced state and resolve imbalances by adjusting their attitudes toward others. The balance theory of celebrity persuasion assumes a triadic relationship, where P is the receiver, O the communicator, and X is the endorsement target. In addition, the dynamics of these triadic relationships depend on the circumstances of connections among the receiver, communicator, and object (Mowen 1980; Woodside and Chebat 2001). In advocacy situations, O positively endorses X; thus, P will decide to accept or reject this persuasion by considering the P–O relationship (Roy et al. 2012; Wood and Herbst 2007). Hence, if individuals positively evaluate an advocate athlete, they are more likely to support their advocacy. On the other hand, negative evaluations of communicators should result in rejection of the target content. This decision-making based on source evaluation is especially pronounced when the recipient is insensitive to the topic of persuasion, indicating when the motivation and ability to process persuasion are relatively low (Petty and Cacioppo 1986; Woodside and Chebat 2001). This theory has been employed to explain the influence of celebrities on marketing (Min et al. 2019; Roy et al. 2012), tourism advertising (Zhang et al. 2020), and political campaigns (Wood and Herbst 2007).

In advocacy appeals, evaluations of communicators are not spontaneously generated; receivers form evaluations from multiple pieces of information associated with the message. The present study utilizes attribution theory to understand the formation of the public's evaluations of athlete advocacy (Heider 1958; Kelley 1973; Malle 2011). Attribution theory explains how people make causal inferences about others' actions and has been applied in the context of celebrities' social activities (Park and Cho 2015; Garcia de los Salmones et al. 2013). This theory posits that people tend to consider the motivations underlying others' actions (Heider 1958). These attributions are based on the information related to communicators or circumstances (Kelley 1973). In the issue advocacy context, receivers attribute altruistic or egoistic motivations to advocacy efforts based on observations of communicators' actions (Ellen et al. 2000; Garcia de los Salmones et al. 2013; Park and Cho 2015). Researchers have reported that altruistic-motivated actions are perceived as favorable, whereas egoistic-motivated efforts are evaluated as unfavorable (e.g., Ellen et al. 2006; Park and Cho 2015). In addition, these attributions could eventually lead to a

communicator's evaluation (Heider 1958). Thus, how receivers perceive athlete advocacy can be an important cue through which receivers evaluate athletes.

From the receiver perspective, the present study examines what factors may increase or decrease the public's issue involvement in athlete advocacy. Through the lens of balance theory, we apply perceived credibility and hypocrisy as source evaluations of advocate athletes. Using attribution theory, we employ perceived fit, effort expended, and role model status as antecedents of communicator evaluations. To explain the individual's involvement in social issues, we develop a two-layered hypothesized model for the relationships among these concepts (Figure 1). In the following subsections, we present our hypotheses.

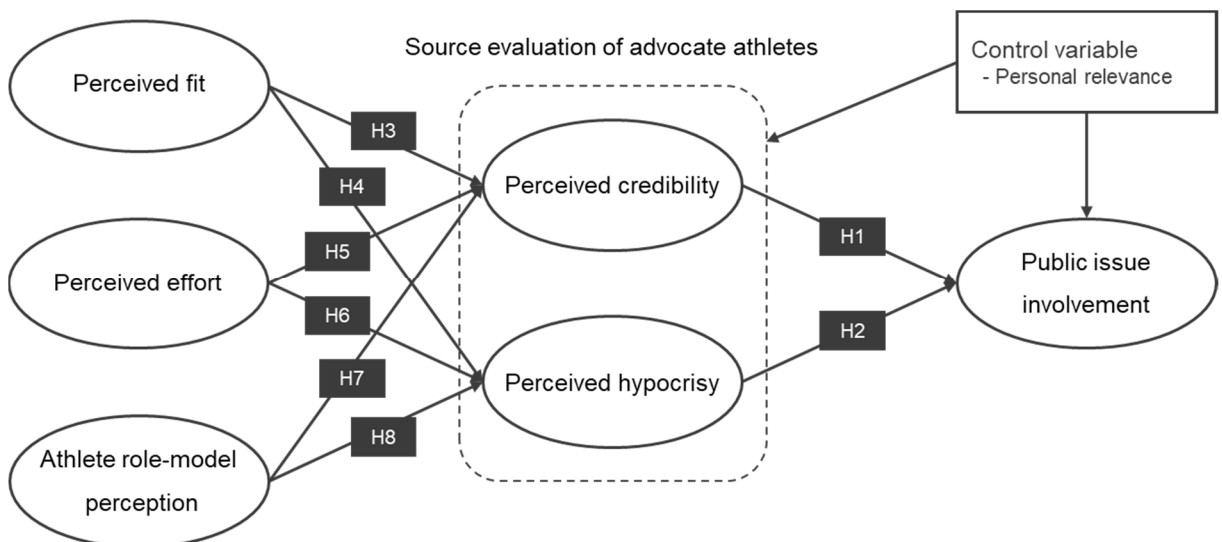

**Figure 1.** Hypothesized Model. H = hypothesis.

### 2.3. Effects of Source Evaluation on Public Issue Involvement

2.3.1. The Role of Perceived Credibility

While communicating social issues, how receivers evaluate the communicator determines the persuasion effectiveness (Hovland et al. 1953; Ohanian 1990). Researchers have reported various aspects of source characteristics, such as source credibility, attractiveness, similarity to recipient identity, and performance (e.g., Amos et al. 2008; Halder et al. 2021; Hovland et al. 1953). In particular, scholars focusing on persuasion aiming at individuals' attitudinal and behavioral changes have discussed the role of source credibility (Hovland et al. 1953; Inoue and Kent 2014). Source credibility refers to the extent to which a "source is perceived as possessing expertise relevant to the communication topic and can be trusted to give an objective opinion on the subject" (Goldsmith et al. 2000, p. 43). This definition posits two crucial dimensions of source credibility: trustworthiness and expertise (Hovland et al. 1953). Trustworthiness is defined as the perceived willingness of the source to make valid assertions, whereas expertise refers to the perceived ability of the source to make valid assertions (McCracken 1989). In other words, the perceived credibility of an advocacy appeal is interpreted as the receivers' evaluations of the communicators' credibility as an information source.

The role of perceived credibility of message sources in persuasion aiming for attitude change has been extensively examined (Amos et al. 2008; Halder et al. 2021; Hovland et al. 1953). Previous researchers have studied perceived credibility in the context of celebrity endorsements for nonprofit and political campaigns (e.g., Jackson and Darrow 2005; Nisbett and DeWalt 2016). Most source credibility studies state that persuasion is more likely to be accepted when the communicator is perceived as a credible source (Amos et al. 2008; Halder et al. 2021). This element of the source credibility model can be explained by balance theory (Heider 1946). In advocacy appeals, perceiving an advocate athlete as credible and,

thus, formulating a positive attitude toward them increases the likelihood that individuals will support the advocated position. Thus, the following hypothesis is developed:

**Hypothesis 1 (H1).** *Perceived credibility of advocate athletes will be positively associated with public involvement in social issues.*

2.3.2. The Role of Perceived Hypocrisy

In issue advocacy, the ethical evaluation of sources plays a key role in persuasion situations. In particular, as celebrity engagements in social issues are often regarded as hypocritical efforts, the role of perceived hypocrisy is also noteworthy (Robeers and Van den Bulck 2021). Hypocrisy refers to "the morally discrediting interpretation of perceived word-indeed misalignment" (Effron et al. 2018, p. 65). People have a sense of hypocrisy not only regarding themselves but also others (Barden et al. 2005). Wagner et al. (2009) conceptualized perceived hypocrisy as consumers' evaluations of an organization's pro-social communication. In addition, messages about engagement in social issues can paradoxically provide an opportunity for the public to generate hypocritical perceptions (Waddock and Googins 2011). This paradox is due to the emphasis on the additional benefits derived from social issue advocacy. Engagement in social issues could be perceived as a vehicle for athletes' self-promotion and, thus, attracts criticism of egocentrism rather than moral motives (Samman et al. 2009). Hence, in the context of engagement in social issues, the sense of hypocrisy can be interpreted as reflecting the public's ethical judgment of athletes' issue advocacy (Shim and Yang 2016). These ethical evaluations of sources can lead to negative consequences in the aspects of the receivers' attitudes (Shim and Yang 2016; Wagner et al. 2009) and behaviors (Jain et al. 2021; Zhigang et al. 2020).

According to balance theory, negative attitudes toward advocacy communicators negatively influence persuasion effectiveness (Heider 1946). Wagner et al. (2009) found that perceived hypocrisy of corporate social responsibility (CSR) messages can have negative consequences in cause-related communication. In addition, celebrity engagement can cause the audience's skepticism, and this perception becomes an important requirement that can spill over to the acceptability of the message in persuasion situations (Jain et al. 2021; Robeers and Van den Bulck 2021; Samman et al. 2009). Thus, perceived hypocrisy in athlete advocacy can hurt the athletes' ability to encourage involvement in social issues.

**Hypothesis 2 (H2).** *Perceived hypocrisy of advocate athletes will be negatively associated with public involvement in social issues.*

*2.4. Antecedents of Source Evaluation in Athlete Advocacy*

2.4.1. Perceived Fit and Effort

Public perceptions of advocacy-related attributes can influence their evaluations of the athletes (Haley 1996; Inoue and Kent 2014). Ellen et al. (2006) and Du et al. (2010) suggested that such attributes, including perceived communicator–cause fit and communicator effort, affect the receivers' responses to the message. Perceived fit is commonly defined in celebrity endorsement research as "the similarity or consistency between the brand and the celebrity" (Bergkvist et al. 2016, p. 173). Empirical studies argue that when people perceive the communicator–cause relationship as congruent, they also perceive altruism, leading to positive evaluations of the communicator (Bergkvist et al. 2016; Garcia de los Salmones et al. 2013; Ilicic and Baxter 2014; Park and Cho 2015). Thus, those who positively evaluate athlete–cause fit are likely to form positive evaluations of advocate athletes. Hence, the following hypotheses are proposed:

**Hypothesis 3 (H3).** *Perceived fit will be positively associated with the perceived credibility of advocate athletes.*

**Hypothesis 4 (H4).** *Perceived fit will be negatively associated with the perceived hypocrisy of advocate athletes.*

Meanwhile, perceived effort refers to "the amount of energy put into a behavior" (Mohr and Bitner 1995, p. 243). This concept can be defined as the amount of energy an athlete puts into an advocacy appeal. Ellen et al. (2000) suggested that people perceive the sender as more generous and caring if they believe the sender is expending more effort on advocacy; Du et al. (2010) suggested a similar effect for companies. Conversely, low-effort advocacy may be regarded as extrinsically motivated behavior intended to create a positive impression, consequently inducing unfavorable evaluations (Lim 2013; Noland 2020). In summary, perceptions of the efforts athletes devote to advocacy influence evaluations of the athletes. Thus, we propose the following hypotheses:

**Hypothesis 5 (H5).** *Perceived effort will be positively associated with the perceived credibility of advocate athletes.*

**Hypothesis 6 (H6).** *Perceived effort will be negatively associated with the perceived hypocrisy of advocate athletes.*

2.4.2. Athlete Role Model Perception

Whether receivers perceive athletes as role models may affect evaluation and persuasion outcomes, as athletes are often considered objects of social admiration. Such perceptions can be understood as general evaluations of athletes that function as bases for evaluations of their advocacy. Haley (1996) found that communicators' general attributes are antecedents of evaluations of their advocacy messages. Moreover, several studies (e.g., Bandura 2001; Bush et al. 2004; Dix et al. 2010) have argued that endorsements by role model athletes can encourage supportive behavior and attitudes toward endorsement objects. These arguments suggest that evaluations of advocate athletes are influenced by how favorable the athletes are viewed as role models. Therefore, the following hypotheses are formulated:

**Hypothesis 7 (H7).** *Perception of advocate athletes as role models will be positively associated with the perceived credibility of advocate athletes.*

**Hypothesis 8 (H8).** *Perception of advocate athletes as role models will be negatively associated with the perceived hypocrisy of advocate athletes.*

*2.5. Control Variable*

In addition to the hypothesized effects, other variables may affect the evaluation of advocate athletes and issue involvement. For the hypothesis testing, this study applies a control variable: personal issue relevance. Personal issue relevance, which refers to the degree to which a particular issue is relevant to an individual's life, may also influence the public's involvement in an issue and evaluation of advocate athletes (Petty and Cacioppo 1986). This concept appears when people predict that an issue "will have important consequences for their own lives" (Apsler and Sears 1968, p. 162). The elaboration likelihood model (Petty and Cacioppo 1986), one of the prime theories explaining the effectiveness of persuasion, posits that the relevance of an issue to an individual can cause a disturbance in the effectiveness of the message. Based on this discussion, we conceptualized the personal issue relevance as a control variable to take into account their potential effects on the hypothesized relationships.

**3. Research Context**

The present study is contextualized in Japanese responses to Naomi Osaka's racial advocacy. Osaka is a leading athlete who has explicitly shown support for BLM, a movement against racism. She has continuously engaged in outreach on racial issues. One of

her most symbolic activities was her declaration that she intended to boycott the Western and Southern Open, which drew considerable attention. Furthermore, at the US Open in September 2020, she appeared before each of her matches wearing a face mask that featured the names of Black victims of racism. Several of her activities represented a protest against the shooting of Jacob Blake, a Black man, by a white police officer on 23 August 2020. Along with these protests, Osaka has spread her message through social media platforms. Her advocacy has received considerable media coverage in Japan, which is the research setting for this study.

Although racism and racial discrimination are becoming increasingly prevalent and visible in Japan, the Japanese people are generally considered insensitive to racism in their country (Iwabuchi and Takezawa 2015). Additionally, not many celebrities address political issues in Japan and other Asian nations (Deflem 2022b). Given this cultural background, it is believed that Osaka's message highlighted her belief and motivated people to be aware of racial discrimination (Deflem 2022a; Reid 2020). Consistent with the purpose of general advocacy, her messages were intended to enhance public involvement in racial issues, such as awareness, concern, knowledge, and information-seeking behavior (Brockington 2014; Ramsay 2020). These outcomes of interest are supposed to be initial steps for the Japanese, who have been relatively insensitive to racial discrimination, to think about racial issues.

## 4. Method

### 4.1. Participants and Procedure

4.1.1. Questionnaire Development and Pilot Study

To ensure the clarity of survey instruments, we began with an assessment of content and face validity using a panel of experts and a pilot study. First, our measurement tool was examined by two academic experts and eight postgraduate students in sport management for face validity, the wording in the local language, and suitability to the athlete advocacy context. Based on this panel's suggestions, we made minor changes to the item wording. Then, a pilot study was conducted to assess the instrument's readability and validity with 32 undergraduate students who were enrolled in sport management courses at a Japanese university. As a result, no concerns or issues arose in the pilot study. Through this process, the final version of the questionnaire included 23 items.

4.1.2. Main Study

For the main study, data were collected through an internet-based survey conducted by a Japanese online survey firm, executed independently of the pilot test. Participants were stratified into 12 groups by gender (female/male) and age (18–24/25–34/35–44/45–54/55–64/65–74 years). Respondents were selected in a manner to ensure that an even sample size was achieved. Potential participants received an invitation from the survey firm noting that they were eligible for a survey. To minimize self-selection bias, the survey was named "questionnaire about yourself". Participants were compensated for their participation by the survey firm.

Overall, 2834 respondents answered the first question, which concerned whether they were aware of Osaka's advocacy. We included respondents who reported awareness of Osaka's advocacy in this analysis to achieve the research purpose. Furthermore, to reduce the risk of respondents providing insincere responses, some items included an "I do not know" response option; we excluded respondents who answered, "I do not know", which further assured that the results were representative of "aware" subjects. Then, those who answered incorrectly to the screening question (i.e., "Please answer X to this question") in the questionnaire and those whose responses were extremely consistent (i.e., those who answered the same anchor for most questions) were excluded from the data analysis. This procedure led to the final sample size of 855; the data did not include any missing values.

To ensure the validity of the cross-sectional study, we randomly divided the sample of 855 participants into test and validation samples: sample A (n = 428) and sample B (n = 427), respectively. The sample size exceeded the minimum recommended size of 161

based on the item/variable ratio (Westland 2010). We then assessed the homogeneity of the two samples by conducting chi-square tests and a t-test for personal attributes (Table 1). No differences were shown between the groups regarding any of the characteristics.

**Table 1.** Sample characteristics.

| Variables | Description | Sample A | | Sample B | | Difference Test |
|---|---|---|---|---|---|---|
| | | n | % | n | % | |
| Gender | Female | 236 | 55.1 | 215 | 50.4 | 1.97 [a, n.s.] |
| | Male | 192 | 44.9 | 212 | 49.6 | |
| Age | Average age | 49.6 | | 49.4 | | 0.16 [b, n.s.] |
| Employment | Full-time | 183 | 42.8 | 177 | 41.5 | 0.15 [a, n.s.] |
| | Other | 245 | 57.2 | 250 | 58.5 | |
| Education | Four-year university degree or more | 216 | 50.5 | 228 | 53.4 | 0.73 [a, n.s.] |
| | Other | 212 | 49.5 | 199 | 46.6 | |
| Marital status | Married | 274 | 64.0 | 286 | 67.0 | 0.82 [a, n.s.] |
| | Other | 154 | 36.0 | 141 | 33.0 | |
| Income | Less than 4 million JPY | 155 | 36.2 | 156 | 36.5 | 5.05 [a, n.s.] |
| | 4–8 million JPY | 171 | 40.0 | 144 | 33.7 | |
| | More than 8 million JPY | 102 | 23.8 | 127 | 29.7 | |

Note. JPY = Japanese Yen; there were no differences for any characteristics difference between sample A (n = 428) and sample B (n = 427); [n.s.] not significant; [a] $\chi^2$-value, [b] *t*-value.

### 4.2. Measures

This study examined how athlete advocacy encourages the public to become involved in racial issues; this was performed by analyzing the relationships among multiple sociopsychological constructs: public issue involvement, personal issue relevance, and perceptions regarding athletes' credibility, hypocrisy, cause fit, effort expended, and role model status. Except for perceived credibility, all variables were measured using a seven-point Likert scale ranging from 1 ("strongly disagree") to 7 ("strongly agree"); perceived credibility was assessed using a seven-point semantic differential scale. All items are listed in Table 2.

We constructed a four-item scale to assess public issue involvement, indicating individuals' cognitive, affective, and behavioral engagements in racial issues. The items were extracted from empirical (Austin et al. 2008; Becker 2012; Casey et al. 2003; Nownes 2021) and descriptive (Babiak et al. 2012; Yan et al. 2018) studies of celebrity and athlete advocacy, reflecting the following aspects: awareness, concern, knowledge, and information-seeking.

As the concepts represent the receivers' evaluations of advocate athletes, we adopted perceived credibility and hypocrisy. Perceived credibility was measured using a six-item scale modified from Ohanian (1990) that focused on receivers' evaluations of celebrity credibility. These items were used in previous athlete endorsement research (Sato et al. 2019). To measure public perception of athlete hypocrisy, we adopted three items from Wagner et al. (2009); given that these items were initially developed in the CSR context, we modified them to reflect athlete advocacy.

The perceived fit between Osaka and her advocacy was measured using two items sourced from the literature (Kim and Na 2007). Perceived effort was measured using two items developed by Mohr and Bitner (1995). Athlete role model perception was measured using a five-item scale developed by Funahashi et al. (2015). As this scale was developed for a general Japanese athlete setting, to reflect the present research context, we modified the wording of items, changing "Japanese athletes" to "Naomi Osaka".

Finally, we included a control variable: personal issue relevance. Personal issue relevance was measured using a single item developed through consideration of previous studies (Inoue and Kent 2012; Apsler and Sears 1968).

**Table 2.** Standardized factor loadings (λ), average variance extracted (AVE), and composite reliability (CR) for the measurement model (Sample A: n = 428).

| Constructs | Items | λ | AVE | CR |
|---|---|---|---|---|
| Perceived credibility (PC) | PC1. Dishonest/Honest | 0.77 | 0.71 | 0.94 |
| | PC2. Insincere/Sincere | 0.80 | | |
| | PC3. Untrustworthy/Trustworthy | 0.87 | | |
| | PC4. Not an expert/Expert | 0.82 | | |
| | PC5. Inexperienced/Experienced | 0.88 | | |
| | PC6. Unqualified/Qualified | 0.92 | | |
| Perceived hypocrisy (PH) | PH1. Naomi Osaka acts hypocritically. | 0.88 | 0.72 | 0.88 |
| | PH2. What Naomi Osaka says and does are two different things. | 0.77 | | |
| | PH3. Naomi Osaka pretends to be someone that she is not. | 0.88 | | |
| Perceived fit (PF) | PF1. Naomi Osaka and the BLM movement fit together well. | 0.93 | 0.71 | 0.83 |
| | PF2. Naomi Osaka and the BLM movement have a lot of similarities. | 0.75 | | |
| Perceived effort (PE) | PE1. Naomi Osaka puts a lot of effort into her advocacy. | 0.91 | 0.79 | 0.88 |
| | PE2. Naomi Osaka spends much time in her advocacy. | 0.87 | | |
| Athlete role model perception (ARM) | ARM1. Naomi Osaka provides a good model for me to follow. | 0.93 | 0.84 | 0.96 |
| | ARM2. Naomi Osaka leads by example. | 0.89 | | |
| | ARM3. Naomi Osaka sets a positive example for others to follow. | 0.90 | | |
| | ARM4. Naomi Osaka exhibits the kind of work ethic and behavior that I try to imitate. | 0.92 | | |
| | ARM5. Naomi Osaka acts as a role model for me. | 0.93 | | |
| Public issue involvement (PI) | PI1. I am aware of the seriousness of racial discrimination. | 0.84 | 0.62 | 0.87 |
| | PI2. I am interested in racial discrimination. | 0.90 | | |
| | PI3. I know a lot about racial discrimination. | 0.70 | | |
| | PI4. I actively seek out information concerning racial discrimination. | 0.69 | | |
| Personal relevance (PR) | PR1. The condition of racial discrimination affects the quality of my life. | - | - | - |

Note. Measurement model fit: $\chi^2/df$ = 3.66 (769.30/210), $p < 0.001$; CFI = 0.94; TLI = 0.92; RMSEA = 0.079; SRMR = 0.049; all standardized factor loadings (λ) were significant ($p < 0.001$).

### 4.3. Data Analysis

For the preliminary analysis, we first examined the normality of the data using the Skewness–kurtosis test (Table A1). The skewness values were less than 3 (from −0.78 to 0.67), and the kurtosis values were less than 7 (from −0.51 to 1.06), indicating that the data were normally distributed (Kim 2013).

Before testing the proposed hypotheses, a confirmatory factor analysis (CFA) was performed using Sample A to assess the adequacy of all constructs. The appropriateness of the measurement model was assessed using the criteria of the overall model fit index, composite reliability (CR), convergent validity, and discriminant validity. We defined the cut-off values for the fit indices as follows: $\chi^2/df$: $\leq 5$, confirmatory fit index (CFI) and Tucker–Lewis index (TLI): $\geq 0.90$, root mean square error of approximation (RMSEA): $\leq 0.08$, and standardized root mean square residual (SRMR): $\leq 0.10$ (Hair et al. 2010; Kline 2016). We also tested the replicability of the measurement model using another CFA with Sample B. Then, we employed structural equation modeling (SEM) to examine the hypothesized relationships. The structural model's overall fit was assessed using the same indices and criteria ($\chi^2/df$, CFI, TLI, RMSEA, and SRMR) as the measurement model test.

### 4.4. Common Method Variance

As the independent and dependent variables were taken from the same sample, it was necessary to consider the potential impact of common method variance (CMV). We applied the following procedures to minimize CMV: (1) notifying respondents that their responses were anonymous and that the data would be treated confidentially; (2) informing respondents that there were no right or wrong answers; and (3) randomizing the order of some

scale items to control for possible item order effects. Furthermore, we employed Harman's single-factor test as a post hoc statistical procedure. An exploratory factor analysis using the unrotated principal axis factoring method was conducted on all psychological measurements (k = 23); multiple factors were identified. The value of the Kaiser–Meyer–Olkin measure was 0.93, indicating that it was an adequate sample size for the factor analysis. Bartlett's test of sphericity ($\chi^2$ = 16333.61, *df* = 253, *p* < 0.001) indicated that the sample was suitable for the factor analysis. The first factor accounted for 44.87% of the total variance, below the criterion value of 50%, indicating that no dominant factor was identified. These results indicated that CMV bias was not a serious problem affecting the validity of this study (Podsakoff et al. 2003).

## 5. Results

### 5.1. Measurement Model

To assess the measurement model, CFA with maximum likelihood estimation was conducted using Sample A (Table 2). The goodness-of-fit indices indicated that the overall fit of the measurement model was good ($\chi^2/df$ = 769.30/210 = 3.66, *p* < 0.001, CFI = 0.94, TLI = 0.92, RMSEA = 0.079, SRMR = 0.049). We then examined the reliability, convergent validity, and discriminant validity (Tables 2 and 3). The composite reliability (CR) values for all seven constructs ranged from 0.83 to 0.96, exceeding the threshold of 0.70 (MacKenzie et al. 2011). All factor loadings were within the acceptable range (0.70–0.93). The average variance extracted (AVE) for all constructs (0.62–0.84) exceeded 0.50, indicating acceptable convergent validity (Hair et al. 2010). All correlations were less than the cut-off point of 0.85 (Kline 2016). Across all pairs of constructs, the square root of the AVE was greater than the correlation coefficient (Fornell and Larcker 1981). The results showed acceptable internal consistency, convergent validity, and discriminant validity.

**Table 3.** Descriptive statistics and correlations of the constructs (Sample A: n = 428).

| Constructs | Mean | SD | Correlation Matrix | | | | | | |
|---|---|---|---|---|---|---|---|---|---|
| | | | **1** | **2** | **3** | **4** | **5** | **6** | **7** |
| 1. Perceived credibility | 5.03 | 1.14 | *0.85* | | | | | | |
| 2. Perceived hypocrisy | 2.60 | 1.12 | −0.67 *** | *0.85* | | | | | |
| 3. Perceived fit | 4.77 | 1.04 | 0.73 *** | −0.60 *** | *0.84* | | | | |
| 4. Perceived effort | 4.78 | 1.00 | 0.52 *** | −0.48 *** | 0.72 *** | *0.89* | | | |
| 5. Athlete role model perception | 4.46 | 1.34 | 0.75 *** | −0.58 *** | 0.62 *** | 0.42 *** | *0.92* | | |
| 6. Public issue involvement | 3.97 | 1.06 | 0.38 *** | −0.36 *** | 0.39 *** | 0.40 *** | 0.51 *** | *0.79* | |
| 7. Personal relevance | 3.48 | 1.53 | 0.15 ** | −0.06 | 0.11 * | 0.15 ** | 0.35 *** | 0.47 *** | - |

Note. The diagonal (in bold and italics) shows the square root of AVE for each construct; * *p* < 0.05, ** *p* < 0.01, *** *p* < 0.001.

To assess the replicability of the measurement model constructed in the previous CFA using sample A, another CFA was conducted using sample B (n = 427). The fit of the measurement model was satisfactory ($\chi^2/df$ = 712.62/210 = 3.39, *p* < 0.001, CFI = 0.93, TLI = 0.92, RMSEA = 0.075, SRMR = 0.052). These results suggest that the measurement model exhibits replicability for different samples.

### 5.2. Structural Model

For hypothesis testing, SEM with maximum likelihood estimation was performed using Sample B (Figure 1). A control variable (i.e., personal relevance) was included to examine the hypothesized relationships more rigorously. The goodness-of-fit of the structural model was acceptable ($\chi^2/df$ = 801.46/214 = 3.75, *p* < 0.001, CFI = 0.92, TLI = 0.91, RMSEA = 0.080, SRMR = 0.066). Table 4 provides the results of the hypothesized path, followed by the effect of the control variable on the receivers' evaluations of the advocate athlete (i.e., perceived credibility and hypocrisy) and advocacy outcome (i.e.,

public issue involvement). The structural model accounted for 64.5% of the variance in perceived credibility, 29.0% in perceived hypocrisy, and 32.8% in public involvement in racial issues.

**Table 4.** Standardized results of the structural model (Sample B: n = 427).

|  | Antecedents | Consequences | $\beta$ | SE | Hypothesis |
|---|---|---|---|---|---|
| **Hypothesized direct effects** | | | | | |
| H1 | Perceived credibility | Public issue involvement | 0.29 *** | 0.05 | Supported |
| H2 | Perceived hypocrisy | Public issue involvement | −0.18 *** | 0.08 | Supported |
| H3 | Perceived fit | Perceived credibility | 0.39 *** | 0.05 | Supported |
| H4 | Perceived fit | Perceived hypocrisy | −0.20 * | 0.08 | Supported |
| H5 | Perceived effort | Perceived credibility | 0.03 | 0.05 | Rejected |
| H6 | Perceived effort | Perceived hypocrisy | 0.07 | 0.09 | Rejected |
| H7 | Athlete role model perception | Perceived credibility | 0.48 *** | 0.04 | Supported |
| H8 | Athlete role model perception | Perceived hypocrisy | −0.45 *** | 0.06 | Supported |
| **Effects of control variable on athlete evaluations and advocacy outcome** | | | | | |
|  |  | Perceived credibility | −0.01 | 0.02 | |
|  | Personal relevance | Perceived hypocrisy | 0.07 | 0.04 | |
|  |  | Public issue involvement | 0.37 *** | 0.03 | |
| **Squared Multiple Correlations ($R^2$)** | | | | | |
|  | Perceived credibility | | 0.65 | | |
|  | Perceived hypocrisy | | 0.29 | | |
|  | Public issue involvement | | 0.33 | | |

Note. Structural model fit: $\chi^2/df$ = 3.75 (801.46/214), $p < 0.001$; CFI = 0.92; TLI = 0.91; RMSEA = 0.080; SRMR = 0.066; $\beta$ = Standardized coefficients; * $p < 0.05$; *** $p < 0.001$.

Perceived credibility was positively associated with public involvement in racial issues ($\beta_{H1}$ = 0.29, $p < 0.001$), supporting H1. Perceived hypocrisy had a significant negative association with public involvement ($\beta_{H2}$ = −0.18, $p < 0.001$), confirming H2. Perceived fit was significantly associated with perceived credibility ($\beta_{H3}$ = 0.39, $p < 0.001$) and hypocrisy ($\beta_{H4}$ = −0.20, $p = 0.014$), confirming H3 and H4. However, perceived effort was not a significant predictor of perceived credibility ($\beta_{H5}$ = 0.03, $p = 0.54$) or perceived hypocrisy ($\beta_{H6}$ = 0.07, $p = 0.32$), rejecting H5 and H6. Athlete role model perception was significantly associated with perceived credibility ($\beta_{H7}$ = 0.48, $p < 0.001$) and perceived hypocrisy ($\beta_{H8}$ = −0.45, $p < 0.001$), supporting H7 and H8.

Finally, we examined the effects of the control variable on source evaluations and the advocacy outcome. Personal relevance was considered a control variable that might confound the results of the hypothesized model. The analysis indicated that personal relevance ($\beta$ = 0.37, $p < 0.001$) was significantly associated with public involvement in racial issues.

### 5.3. Mediation Analysis

The proposed model implies that perceived credibility and hypocrisy mediate the effects of perceived fit, perceived effort, and athlete role model perception on public issue involvement (Table 5). We employed bootstrapping estimation with 5000 resamples to calculate bias-corrected 95% confidence intervals (CIs) for indirect effects. This analysis was based on the structural model with the control variable. Perceived fit was significantly related to public involvement in racial issues, mediated by perceived credibility ($\beta$ = 0.12, $p < 0.001$, 95% CI [0.06, 0.22]) and perceived hypocrisy ($\beta$ = 0.04, $p = 0.04$, 95% CI [0.001, 0.11]). Additionally, athlete role model perception was positively associated with advocacy outcomes through perceived credibility ($\beta$ = 0.12, $p = 0.004$, 95% CI [0.07, 0.20]) and hypocrisy ($\beta$ = 0.07, $p = 0.011$, 95% CI [0.02, 0.14]). Conversely, we found no significant indirect effects of perceived effort on public involvement through perceived credibility ($\beta$ = 0.01, $p = 0.54$, 95% CI [−0.04, 0.06]) and hypocrisy ($\beta$ = −0.01, $p = 0.36$, 95% CI [−0.08, 0.02]).

**Table 5.** Results of mediation analyses (Sample B: n = 427).

| Paths | β | SE | 95% CI Lower | 95% CI Upper |
|---|---|---|---|---|
| Perceived fit → Perceived credibility → Public issue involvement | 0.12 *** | 0.07 | 0.06 | 0.22 |
| Perceived fit → Perceived hypocrisy → Public issue involvement | 0.04 * | 0.03 | 0.00 | 0.11 |
| Perceived effort → Perceived credibility → Public issue involvement | 0.01 | 0.05 | −0.04 | 0.06 |
| Perceived effort → Perceived hypocrisy → Public issue involvement | −0.01 | 0.03 | −0.08 | 0.02 |
| Athlete role model perception → Perceived credibility → Public issue involvement | 0.12 ** | 0.04 | 0.07 | 0.20 |
| Athlete role model perception → Perceived hypocrisy → Public issue involvement | 0.07 * | 0.03 | 0.02 | 0.14 |

| Constructs | Direct Effect | Indirect Effect | Total Effect |
|---|---|---|---|
| Perceived credibility | 0.39 | - | 0.39 |
| Perceived hypocrisy | −0.17 | - | −0.17 |
| Perceived fit | - | 0.15 | 0.15 |
| Perceived effort | - | −0.004 | −0.004 |
| Athlete role model perception | - | 0.19 | 0.19 |
| Personal relevance | 0.26 | −0.01 | 0.25 |

Note. β = unstandardized coefficients. CI = confidence interval; * $p < 0.05$; ** $p < 0.01$; *** $p < 0.001$.

## 6. Discussion

Researchers have anecdotally argued that athlete advocacy can drive the public's cognitive, affective, and behavioral involvement in social issues (Babiak et al. 2012; Cunningham et al. 2021; Pelak 2005). Thus, it is important to examine how advocate athletes can develop their persuasiveness. The present study aimed to examine how the evaluations of advocate athletes function in motivating the public's issue involvement, focusing on Naomi Osaka's racial advocacy in the Japanese context.

The present study contributes to the literature on athletes' social influences by answering the question: "How do the evaluations of advocate athletes function in fostering public issue involvement?" To elaborate, the theoretical foundations we adopted in this study (i.e., balance theory and attribution theory) advance our understanding by explaining the persuasive function of source evaluation in athlete advocacy. Using balance theory (Heider 1946; Mowen 1980), we identified the effect of source evaluation (i.e., perceived credibility and hypocrisy) on the social outcomes of athlete advocacy (i.e., public issue involvement). Although most studies (Amos et al. 2008; Halder et al. 2021) in this field have focused on the positive aspects of information sources, this study introduced the concept of source hypocrisy, which is the public's ethics-based evaluations. The observation that perceived hypocrisy functions to enhance persuasiveness similarly to perceived credibility illuminates the role of the public's ethical evaluation in issue advocacy. This finding contributes to establishing a multidimensional view of source evaluations, and theoretically advances the literature on the source effects in issue advocacy (Jain et al. 2021; Samman et al. 2009). From the attribution theory perspective (Heider 1958; Kelley 1973), we examined whether perceptions of athlete–cause fit, effort expended, and role model status are determinants of evaluations of advocate athletes. Our findings highlight the role of interactions among the communicator, receiver, and content and empirically illustrate the functions of these attributes in the context of athletes' issue advocacy (Haley 1996; Inoue and Kent 2014). The findings are further discussed in the following sections by referring to the existing literature.

Consistent with balance theory, we found that perceived credibility is positively associated with public issue involvement, whereas perceived hypocrisy is negatively related. These results indicate that, in interpersonal relationships, perceived credibility and hypocrisy are significantly associated with the outcome of athlete advocacy (i.e., raising the public's cognitive concern, providing issue knowledge, and encouraging information-seeking behavior). This empirical evidence supports the existing literature (Amos et al. 2008; Halder et al. 2021; Ohanian 1990) that argues that perceived credibility is a significant

determinant of persuasion effectiveness. Additionally, the findings provide empirical insights into findings from descriptive studies concerning public skepticism of celebrity engagement (Robeers and Van den Bulck 2021; Samman et al. 2009). This result may be somewhat due to the context of this study. Respondents could have prioritized their perception of the source (i.e., Naomi Osaka) in their acceptance of persuasion because racial issues may be considered as not so critical in Japan's social circumstances (Iwabuchi and Takezawa 2015). Moreover, our findings lead to the idea that Osaka needs to be perceived as credible, knowledgeable, and non-hypocritical to raise public cognitive, emotional, and behavioral engagement in racial issues. Although athletes are not always technical experts on social issues (Sports Philanthropy Project 2011), this idea would be helpful for other athletes who are seeking to promote effective issue advocacy.

This study showed that perceived athlete–cause fit is significantly related to athletes' credibility and hypocrisy. This result indicates that individuals who perceive athlete–cause fit as high tend to evaluate athletes' credibility and hypocrisy positively, supporting previous research on celebrities' involvement in social issues (Bergkvist et al. 2016; Garcia de los Salmones et al. 2013; Ilicic and Baxter 2014). This observation is consistent with attribution theory and can be interpreted as indicating that a strong perceived fit between athletes and racial issues facilitates a positive evaluation of the advocate athletes. It is clear that communicator characteristics, such as race and identity, matter in public perceptions (Deflem 2022a; McCracken 1989). In this study's context, Naomi Osaka highlighted the compatibility of her BLM support with the statement, "before I am an athlete, I am a Black woman". (Jurejko 2020). This communication, which aims to stress the athlete–issue congruence, can be considered a strategy for improving her reputation as an advocate.

Meanwhile, the results also implied that perceptions of the effort athletes devoted to social issues might not be a determinant factor in evaluating athlete advocacy. Although this finding contradicts our hypothesis, it can be interpreted from multiple perspectives. First, the "discounting principle", a psychological mechanism for evaluating and reasoning about others, may provide a better explanation for our results. This principle proposes that when one cause of a phenomenon is emphasized, the perceived influence of other causes diminishes (Kelley 1973). Our findings might reveal a discounting of the role of perceived effort due to the prominence of other causes, such as perceived fit and athlete role model status. Additionally, we should consider the possibility of a backlash against athlete advocacy. Generally, the amount of effort expended is positively related to the effectiveness of messages (Du et al. 2010); however, athletes who engage excessively in racial issues may receive public criticism and institutional sanctions (Kaufman 2008). Indeed, after Osaka expressed her support for BLM by declaring her intention to withdraw from a tournament, she faced not only praise but also considerable criticism for bringing politics into sport (McNeil 2020). This complicated structure regarding athletes' advocacy may distort people's evaluations of advocates.

Finally, this study revealed that the receiver's perception of an advocate athlete's role model status determines perceived credibility and hypocrisy and is indirectly related to involvement in racial issues. These results provide empirical support for role models' influence in observational learning (Bandura 2001). The finding that whether athletes are perceived as positive role models indirectly determines the effectiveness of issue advocacy extends our understanding of athletes' social influences (Bush et al. 2004; Dix et al. 2010). As celebrities in Japan have not often engaged in advocacy and activism, some participants might have recognized Osaka as a notable advocate due to her role model attributes, such as the uniqueness of her resistance to racism besides on-the-field successes (Deflem 2022b). Meanwhile, for participants who do not value issue engagement in the role model element of athletes, Osaka's advocacy effort can be a potential cause for discomfort, which leads to a negative evaluation. Given the association between role model status and source evaluation, future research should focus on how athletes become perceived as positive role models.

This study has several implications for athletes who engage in advocacy and seek to facilitate their fame as a vehicle for promoting social and political causes. A point worth noting is that not all athletes address political and institutional issues. Furthermore, "athletes should not be required—or even expected—to take public stances on issues important to them and/or their communities" (Coombs and Cassilo 2017, p. 439). However, athletes who decide to advocate social issues can be agents for motivating the public to be involved in social and political issues (Agyemang et al. 2020; Kaufman 2008; Pelak 2005). Therefore, a discussion on how advocate athletes can enhance the persuasiveness of their messages is needed. The current study showed that the outcome of advocacy is related to the receivers' evaluations of advocate athletes. In addition, our data revealed that advocacy appeals that lack of perceived congruence between the athlete and the issue can promote public skepticism, resulting in negative judgments of the advocacy effort. Hence, athletes should carefully consider the social issues they involve themselves in. Moreover, establishing athletes as role models is vital for communicating social issues. Our findings suggest that athletes should aim to be regarded as good role models in society to enhance their effectiveness in advocating issues. Our data imply that deviant behavior that damages athletes' images can hinder the success of their advocacy. This observation indicates that athletes must seek to understand who perceives them as role models and how.

## 7. Conclusions

This study provides evidence on the function of source evaluation in fostering public issue involvement through the lens of balance theory and attribution theory. This study showed that the perceived credibility and hypocrisy of advocate athletes are directly related to individuals' involvement in racial issues. Additionally, these public evaluations of athlete advocacy could be influenced by the athletes–cause fit and their perceptions as role models. The findings of this study enrich the literature on the management of sports for social change by demonstrating the importance of source evaluation in athlete advocacy in achieving advocacy outcomes. Further, these findings may help athletes seeking to fight against social issues develop an effective strategy for promoting issue advocacy.

This study contained several limitations. First, we focused on one case of athlete advocacy (Naomi Osaka and racial issues); thus, although this study provides a unique contribution to clarifying the Japanese response to athlete advocacy, caution should be exercised regarding generalizing the results. Future research could investigate the function of source evaluation in athlete advocacy across different settings and issues. Second, as self-reported questions were used to identify respondents who were aware of Osaka's advocacy, we could not consider the channel through which respondents received this information. As consumers respond differently to athlete advocacy depending on media framing (Park et al. 2020), further investigation of message channels is necessary. Third, as this study could not address actual behavior, there remains space for a better understanding of the link to individuals' political participation, such as donations, volunteering, and social activism. Fourth, this study focused on specific source evaluations of advocate athletes (i.e., credibility and hypocrisy). However, the existing literature (Amos et al. 2008; Hovland et al. 1953) indicates that the persuasiveness of celebrities' messages is associated with other persuasion components (e.g., source attractiveness, on-the-field performance, and message content). Therefore, future research could investigate the effects of diverse persuasion components to fully understand the dynamics of athlete advocacy. Finally, although this study used balance theory and attribution theory to understand the mechanism of influence of athlete advocacy, this result does not fully identify causal relationships. In other words, we cannot deny the causal relationship that individuals' strong involvement in racial issues increases the reputation of advocate athletes. As Knoll and Matthes (2017) highlighted, the effectiveness of celebrity endorsements remains controversial; therefore, future research should examine the effectiveness and dynamics of athlete advocacy using methods that provide greater internal validity.

**Author Contributions:** Conceptualization, W.O.; methodology, W.O.; formal analysis, W.O.; investigation, W.O., H.F. and Y.M.; data curation, W.O.; writing—original draft preparation, W.O.; writing—review and editing, W.O., H.F. and Y.M.; supervision, H.F. and Y.M.; funding acquisition, W.O., H.F. and Y.M. All authors have read and agreed to the published version of the manuscript.

**Funding:** This work was supported by JSPS KAKENHI Grant Number 21K11402 and JST SPRING Grant Number JPMJSP2128.

**Institutional Review Board Statement:** Ethical review and approval were waived for this study since the survey methodology fully guaranteed the anonymity of the participants.

**Informed Consent Statement:** Informed consent was obtained from all subjects involved in the study.

**Data Availability Statement:** The data presented in this study are available on request from the corresponding author.

**Conflicts of Interest:** The authors declare no conflict of interest. The funders had no role in the design of the study; in the collection, analyses, or interpretation of data; in the writing of the manuscript, or in the decision to publish the results.

## Appendix A

**Table A1.** Descriptive statistics (all sample: n = 855).

| Constructs | Items | Mean | SD | Skewness | Kurtosis |
|---|---|---|---|---|---|
| Perceived credibility (PC) | PC1 | 5.47 | 1.21 | −0.76 | 0.92 |
| | PC2 | 5.39 | 1.21 | −0.68 | 0.54 |
| | PC3 | 5.24 | 1.29 | −0.78 | 0.75 |
| | PC4 | 4.50 | 1.28 | −0.18 | 0.25 |
| | PC5 | 4.73 | 1.26 | −0.37 | 0.42 |
| | PC6 | 5.15 | 1.28 | −0.51 | 0.23 |
| Perceived hypocrisy (PH) | PH1 | 2.56 | 1.25 | 0.61 | 0.08 |
| | PH2 | 2.73 | 1.31 | 0.67 | 0.26 |
| | PH3 | 2.57 | 1.21 | 0.62 | 0.18 |
| Perceived fit (PF) | PF1 | 4.96 | 1.13 | −0.44 | 0.57 |
| | PF2 | 4.64 | 1.09 | −0.38 | 1.06 |
| Perceived effort (PE) | PE1 | 4.95 | 1.07 | −0.23 | 0.34 |
| | PE2 | 4.59 | 1.02 | 0.01 | 0.49 |
| Athlete role model perception (ARM) | ARM1 | 4.28 | 1.40 | −0.28 | −0.03 |
| | ARM2 | 4.61 | 1.39 | −0.45 | 0.07 |
| | ARM3 | 4.74 | 1.40 | −0.54 | 0.20 |
| | ARM4 | 4.31 | 1.40 | −0.30 | −0.05 |
| | ARM5 | 4.32 | 1.40 | −0.27 | −0.15 |
| Public issue involvement (PI) | PI1 | 4.53 | 1.22 | −0.48 | 0.14 |
| | PI2 | 4.42 | 1.24 | −0.46 | 0.09 |
| | PI3 | 3.69 | 1.24 | −0.06 | −0.18 |
| | PI4 | 3.11 | 1.28 | 0.22 | −0.25 |
| Personal relevance (PR) | PR1 | 3.48 | 1.53 | 0.25 | −0.51 |

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
