# Peer review of "Examining the Role of Source Evaluation in Athlete Advocacy: How Can Advocate Athletes Inspire Public Involvement in Racial Issues?"

_socsci, doi:10.3390/socsci11080372_

Round 1

Reviewer 1 Report

Thank you for the opportunity to review your script which is a significant contribution to the current literature on athlete advocacy pertaining to societal issues and social justice. Your study is original, coherent and comprehensive. I have made brief comments on each section as follow:

Abstract: Clear, informative and succinct.

Introduction: The article context is established with a plethora of seminal and current literature. Please describe or clarify what you mean or what is understood to be a 'sustainable society' (L6, L36 & L99), and reframe your statement 'that athletes are expected to function as agents for dissemination for social and political issues' (L64-65). In its current form this statement is too general and disregards athlete choice to advocate or not. You have introduced and justified your athlete advocate well, provided appropriate background on her, and clearly established the relevance and contribution your research will make from a receivers' evaluations perspective on athlete advocate persuasiveness.  

Literature Review: The theoretical underpinnings of Heider's (1946) balance theory and Heider (1958) and Kelley's (1973) attribution theory are succinct. Perhaps a more current interpretation of these theories would be useful here. The eight hypotheses to test the research question are clearly presented and justified. The remainder of the literature review is well written with outstanding clarity. You have included figures and tables appropriately.

Methods and Results: No comment.

Discussion: Introduced concisely with all aspects included. Consider inserting the statement, “How does the evaluation of advocate athletes encourage individuals to become involved in social issues?” (L247) within your Introduction section to connect the research question more coherently across your article. Back up statements such as 'significantly related to .." (L454 - 455). Avoid introducing new literature sources in the discussion section that have not been previously referred to (Bandura 2001; Coombs & Cassilo 2017). The inclusion of Coombs and Cassilo's findings on athlete advocacy expectation here in the discussion (L489, 490 & 491) need to connect to this important factor previously highlighted in the introduction (Agyemang et al. 2020; 65 Kaufman 2008; Pelak 2005). Provide greater clarity regarding the terms 'athlete-advocate' and 'athlete-advocacy' within the context of this study. Overall, a well argued section for the relevance and possible outcomes of your study, both positive and negative. 

Conclusion: This section is somewhat brief. Clearly state what the study has found, can contribute to current understandings, and any gaps that need further research. You have however, outlined the limitations with suggestions for further study effectively.

General comment:  The manuscript is well written in a concise, detailed, and coherent manner. The complexity of athlete advocacy in persuading and influencing people's views on societal issues is important to acknowledge and to investigate. Through this study you have shed some light on the deeper influencers on effective persuasion from the respondents' view. This is an innovative and interesting perspective which provides empirical evidence to inform the possible impacts on a population, of elite athlete behaviours and communications. 

Reviewer 2 Report

This is a very relevant topic, and it is great that the authors have provided evidence to extend the knowledge-base to better support researchers, practitioners and athletes alike in this space. 

The research question “How does the evaluation of advocate athletes encourage individuals to become involved in social issues?” is clear. Sound rationales were provided for the chosen methodological framework which draws upon balance theory and attribution theory to engage in a critical exploration athlete advocacy. 

With that said, I am left wondering what type of involvement by the receiver were affected as a result of the athlete's advocacy? The 4 questions asked in the "Public issue involvement" section don't really suggest involvement except for PI4. This may be the reason for why I was left still wondering by the end of the manuscript and therefore, your research question still leaves a lot of room for exploration. Can you address this in your discussion by unpacking the spectrum of involvement as defined in your study (i.e. awareness to actively involved) as it relates to your participants?

At times it did feel a little repetitive in the discussion and therefore would there a couple of paragraphs that should be linked back to Osaka and/or her participants for a more in-depth and critical exploration of the issue at hand.
Line 442 - Para beginning "Consistent with balance theory, we found...": How do athletes (like Naomi Osaka) create interpersonal relationships with their fans? Can you unpack your closing sentence of this paragraph with regards to Naomi Osaka? If this is about the effect of Osaka persuading Japanese people in Japan to become involved in "racial justice" what are some of the cultural nuances (if any) that impact "perceived credibility, knowledgeable, and non-hypocritical".
Line 480 - Para beginning "Finally, this study revealed that...": Again, I would consider adding that space should be created for those cultural nuances to evolve in pushing for more understanding around the evolution of athletes as perceived social role models.
These suggestion come as a result of your leading rationale for the chosen context of this study, within a Japanese context. I am by no means an expert in Japanese knowledge, society, however, it would be remiss to not consider some of the cultural nuances that could be considered in the broader critical thinking.

Overall, I'm excited about what is to come in this space. This is an important conversation that should be engaged in as it can impact the many facets of the professional athlete eco-system (i.e. athlete wellbeing, mental health, marketing, management, coaching etc.).

A minor (and very quick fix). Quotation marks for verbatim quotes should be placed around the quote itself and not the authors. See lines 149, 167, 185, 199, 239, 491.

Reviewer 3 Report

Drawing from the balance theory and attribution theory, the author(s) investigated how Osaka’s advocacy is perceived and evaluated by the Japanese people and motivates them to get involved in racial issues. Specifically, the author(s) looked at the roles of credibility and hypocrisy linking with public involvement. The method and result of the study were well written and easy to read, but restructuring/new insights into the intro and discussion could improve the readability and provide a depth narrative. I hope some of my comments below could be useful to enhance the overall quality of the manuscript.

Abstract

The implications written in the abstract were quite vague and simplistic. Please consider detailing or rephrasing to highlight the significance of the study (lines 17-18).

Intro

I found the intro was too repetitive which could be articulated more clearly and written in an organised manner. For instance, the 1st-3rd paragraphs could have been condensed to address the study background and provide keener focus. The ideas could be refined by athletes’ engagement in social justice issues in general and in particular for their advocacy for racial/ethnic minoritised groups, a brief history and examples of athletes’ advocacy, expectations, etc. Then, its impacts on a message receiver’s evaluations of advocate athletes and their involvement could be of specific interest.

Page 2, after line 77, please provide evidence to back up the argument made that the exposure to Osaka’s advocacy for anti-racism may increase in racial/ethnic awareness of the Japanese people. A little more engagement with existing studies outside the field of sport management would help frame your study more clearly and bring ideas together more coherently.

Page 2, line 87 or elsewhere, as the author(s) focus on the persuasive effectiveness of athletes’ social justice advocacy, some discussion and sources regarding the importance and functions of evaluations of advocacy and influences of acceptance of advocative messages communicated by a celebrity/endorser (outside of the sport) could be worthy and appealing (as you did in LR), linking to the strong justifications for conducing this study.

LR

The flow was confusing as there were antecedents to pick up to frame credibility and hypocrisy. Though the impacts of source evaluations could be the focal point of this study, the antecedents are essential elements influencing both mediators at the same time. Consider presenting the review of the relevant literature and hypotheses about the three antecedents of evaluations (perceived fit, effort expended, and role-model status) first and describing the meditating effects to follow.

Page 4, a more in-depth look into various dimensions of source credibility and hypocrisy might help further clarification. For example, past researchers have identified attractiveness, honesty, expertise, and/or truthfulness of the assertions and content communicated by a celebrity endorser to conceptualise source credibility. Please explore the literature on the mediators.

Sources: Inoue, Y., & Kent, A. (2012). Sport teams as promoters of pro-environmental behavior: An empirical study. Journal of Sport Management, 26, 417-432.

Page 4, Is there any existing research examining the roles of perceived credibility and hypocrisy in sport? I wonder whether message receivers’ initial negative responses to athlete advocacy are associated with higher levels of perceived hypocrisy. Perhaps advocacy messages athletes disseminated could be perceived to be less credible and less authentic when characteristics/identities of a message sender are unattractive and dissimilar to those of receivers (Cunningham et al., 2021), but these messages do not necessarily lead to more hypocritic outcomes. Arguably, demographics or actions of a message sender may make athlete advocacy appear hypocritical rather than the content of communication. However, the receivers’ perceptions based upon past experience and knowledge about the advocate athlete might cultivate trust/distrust in message content and sources. Such a lit review and critical appraisal of previous studies is missing upfront. While the lit review was quite threaded through a sound narrative and underpinned the current study, reorganisation of ideas could thus strengthen this section with a more coherent and clear flow. Consider switching H3-8 and H1-2 around.

Methods

Overall, the method section was written clear and rigorous as the author(s) established the appropriate cutoff values to confirm validity and reliability and justified the use of SEM.

Page 6, given the stratified sampling was used, how did the author(s) recruit the study participants, and have you compensated all 2,834 respondents who participated in the initial survey including those who were invited for the pilot testing? I suggest the authors add another subsection to detail the data collection/analysis procedures of this pilot testing separately.

Page 9, again, please clarify whether the responses gathered during pilot study were incorporated into and used in the actual study.

Page 9, the authors ran an EFA, but did not report the results.

Consider reporting reliability score per each measure under the measure subsection.

Results

Page 9, the sample size was sufficient for conducting CFA and SEM, and there was sufficient evidence of construct, convergent, and discriminant validity as shown in Tables 2-3. Corresponding values of each parameter estimate were assessed suggesting acceptable/adequate model fit with their data. While the author(s) applied appropriate measurement and structural model tests and provided relevant fit statistics, I wonder whether they also previously inspected any missing data, outliers, and normality assumptions for testing their hypothesised model. Report values of skewness and kurtosis where appropriate.

Page 11, given the authors indicated in line 411, please report (total and) indirect effects of the analysis to demonstrate the mediational processes and effects.

Discussion

Lines 426-441, the discussion and conclusion did not link back to the literature covered earlier in the manuscript, particularly to balance theory, and each feature of the concepts was brief.

As with insignificant results on this factor, I would consider removing such statements or discuss more in-depth while comparing/contrasting against the previous studies to suggest the effects of efforts advocate athletes make.

Minor points.

Page 1, line 10, “effects”

Page 1, line 32-34, consider switching these lines and previous ones around or starting a new line for a better logical flow.

Page 2, line 53, hereafter try not to humanise “previous studies.” The same applies to line 515.

Page 3, line 137, change between to among.

Page 4 and onwards, when formulating hypotheses, use future tense instead of the present tense.

Page 6, line 242, “discussion”

Page 6, line 258, something is missing. Make a complete sentence.

Page 6, line 260, add “the” before Japanese.

Page 6, consider presenting the subsection 3.1. before the method section and starting from the subsection 3.2.

Page 7, line 298, add public involvement.

Page 11, line 420-421, sources required to back up the statement. The same to the lines 433-434.

Round 2

Reviewer 3 Report

I previously reviewed the script and enjoyed reading the revised version. I think the author(s) did a good job in answering all the reviewers’ comments and concerns raised in the previous round. In the current form, however I would still recommend the authors consider restructuring the lit review section where the hypothesis development and testing could be presented with a better flow. My remaining suggestions listed below are relatively minor which I hoped to contribute to improving the quality of the manuscript.

Abstract

Consider revising the last sentence something like “promote racial justice issues effectively” instead of the current wording.

Intro

2nd para. on page 2, consider revising the last two sentences, “Thus, this study focuses on Osaka…in her advocacy. The present study intends to examine how public evaluations function,” and insert sometime like “particularly how advocate athletes use their persuasiveness to enhance public’s involvement in racial issue,” instead of the current wording.

3rd para. on page 2, add “the” before “sport management and sociology literature.” The same to the 3rd sentence of this para. adding “the” before “sm research.”

4th para on page 2, may be “an opportunity” than “the.”

LR

4th para on page 3, please consider removing “is a family of theories that” as it flows without.

2nd para on page 5, what stands for CSR? Would remove repeating words.

Method

2nd para under the section 4.1.2., please consider alternative word for “a trap question” such as “screening question.”

The subsection 4.3. here or somewhere else like the subsection 5.1.: further to my previous comment re reliability, I understand the author(s) reported the values of CR in Table 2. I showed them, but I meant the author(s) write out them in the texts perhaps providing a range or indicating whether they were greater than the cutoff.

The subsection 4.4. further to my previous comment re EFA, I am sorry I did miss that the author(s) reported the value of the total variance explained. As the author(s) argued, the manuscript was not focused on scale development, thus the use of both EFA and CFA was likely unnecessary. But, the author(s) conducted both and checked for CMV and/or CMB. To ensure the appropriateness of a factor analysis the author(s) might have taken into account whether the sample was suitable for and whether the model could fit the data well. In line with this, I will be happy to see the results of values of KMO (Kaiser–Meyer–Olkin measure) and Bartlett test to see whether they were greater than .5 and less than .05, respectively.

Discussion

Last para on page 12, add “the” in front of “existing literature.”

Minor points

Could benefit from a proofread to clear a number of errors spread through the script (these include issues with proper referencing).
